# Diagnostic efficiency and validity of the DSM-oriented Child Behavior Checklist and Youth Self-Report scales in a clinical sample of Swedish youth

Gudmundur Skarphedinsson[1]*, Håkan Jarbin[2,3], Markus Andersson[2,3], Tord Ivarsson[4]

1 Faculty of Psychology, University of Iceland, Reykjavik, Iceland, 2 Child and Adolescent Psychiatry, Region Halland, Halland, Sweden, 3 Faculty of Medicine, Department of Clinical Sciences Lund, Child and Adolescent Psychiatry, Lund University, Lund, Sweden, 4 Institute of Neuroscience and Physiology, University of Gothenburg, Gothenburg, Sweden

* gskarp@hi.is

**Data Availability Statement:** All relevant data are within the manuscript and its Supporting Information files.

## Abstract

The Child Behavior Checklist (CBCL) and Youth Self-Report (YSR) are widely used measures of psychiatric symptoms and lately also adapted to the DSM. The incremental validity of adding the scales to each other has not been studied. We validated the DSM subscales for affective, anxiety, attention deficit/hyperactivity (ADHD), oppositional defiant (ODD), conduct problems (CD), and obsessive-compulsive disorder (OCD) in consecutively referred child and adolescent psychiatric outpatients (n = 267) against LEAD DSM-IV diagnoses based on the K-SADS-PL and subsequent clinical work-up. Receiver operating characteristic analyses showed that the diagnostic efficiency for most scales were moderate with an area under the curve (AUC) between 0.70 and 0.90 except for CBCL CD, which had high accuracy (AUC>0.90) in line with previous studies showing the acceptable utility of the CBCL DSM scales and the YSR affective, anxiety, and CD scales, while YSR ODD and OCD had low accuracy (AUC<0.70). The findings mostly reveal incremental validity (using logistic regression analyses) for adding the adolescent to the parent version (or vice versa). Youth and parent ratings contributed equally to predict depression and anxiety disorders, while parent ratings were a stronger predictor for ADHD. However, the youth ADHD rating also contributed. Adding young people as informants for ODD and OCD or adding the parent for CD did not improve accuracy. The findings for depression, anxiety disorders, and ADHD support using more than one informant when conducting screening in a clinical context.

## Introduction

Recent systematic reviews report that at any given year approximately 13–25% of youth suffer from mental disorders that cause significant functional impairment in important domains of everyday life such as family, school, and socializing with peers [1, 2]. This brings about high costs and suffering for the individual, family, and society as a whole [3, 4] warranting efficient

**Funding:** HJ, 2008-22893, Stiftelsen Söderström-Königska, https://www.sls.se/vetenskap/sok-anslag/stift.-soderstrom/, NO role in design, data, publishing, preparation MA, 110361, Development and Education (FoUU) within Region Halland, Sweden, https://vardgivare.regionhalland.se/utveckling-forskning/forskning/projektmedel-och-bidrag-for-forskning-och-utveckling/doktorandmedel/, NO role in design, data, publishing, preparation HJ, 133821, Development and Education (FoU) within Region Skåne, Sweden, https://www.skane.se/organisation-politik/forskning/sa-finansierar-vi-forskningen/, NO role in design, data, publishing, preparation.

**Competing interests:** The authors have declared that no competing interests exist.

and effective assessment and treatment for these young people. However, only a small proportion of youth with mental disorders receive adequate treatment [5]. This is especially true for internalizing disorders which are greatly underdiagnosed and undertreated [6–9]. Thus, the need to identify and treat pediatric mental disorders is important and may potentially reduce the risk of impairment, severity, and recurrence of psychopathology in the future [6, 10, 11].

Standardized diagnostic interviews (SDIs) are considered to be the gold standard [12]. Brief continuous psychometric measures are more time-efficient and thus less expensive. They can be suitable for screening or as part of clinical intake procedures to capture a wide range of symptoms in a cost-efficient fashion [13]. However, it is important to evaluate the diagnostic efficiency of screening instruments using representative samples, such as consecutive treatment seeking children [14, 15].

Collecting and combining data from multiple informants (e.g., parents and children) can increase the accuracy of screening and is recommended as informant discrepancy is common [16–21] and particularly for subjective symptoms and behavior outside the home [22, 23].

The Child Behavior Checklist (CBCL) and Youth Self-Report (YSR) are widely used measures of psychiatric symptoms in young people, measuring a range of problem areas [24]. They are often used as part of clinical intake procedures and can screen for psychiatric disorders without any additional cost for the clinic or the family. The syndrome scales of the CBCL/YSR derived by factor analysis have only shown modest concordance with the Diagnostic and Statistical Manual of Mental Disorders (DSM) [25–28]. For instance, as each syndrome scale is related to multiple DSM disorders (e.g., the anxious/depressed component is related to both depressive disorders and anxiety disorders), making it impossible to tease apart whether a child's symptoms are congruent with depressive or anxiety disorders or both [28]. This lack of concordance is suboptimal as treatment options are based on DSM or International Classification of Diseases (ICD) diagnoses.

The authors of the CBCL/YSR have attempted to overcome this limitation by developing DSM-oriented scales (DOSs) based on expert consensus, choosing pre-existing items corresponding with the DSM criteria. This has resulted in the following DOSs: affective, anxiety, somatic, attention deficit/hyperactivity (ADHD), oppositional defiant (ODD), conduct problems (CD), and obsessive-compulsive disorder (OCD). Several studies have investigated the concurrent validity of the DOSs in clinical samples and compared this with syndrome scales. Ebesutani and colleagues [29, 30] showed that DOSs are not superior to the original syndrome scales, while Bellina [31] showed weaker correspondence between the DOS and DSM diagnoses compared with syndrome scales except for the ADHD scale, which outperformed the older attention problems scale. Further, Aebi [32] showed better correspondence between the affective DOS and DSM-IV diagnosis of major depressive disorder than the older syndrome scales. Most studies have reported acceptable correspondence between the affective DOS and a depressive disorder diagnosis [33–36], the anxiety DOS and anxiety disorders [34–36], and the ADHD, ODD, and CD scales and their corresponding disorders [35, 36]. Some evidence also exists for the OCD scale but not in purely clinical samples [37, 38].

However, we are not aware of any existing study that has combined data from multiple informants to evaluate increased accuracy of the DOS. The present study addresses the lack of data on the diagnostic efficiency of the DOS when combining data from multiple informants. The aim of the current study was to evaluate the concurrent and discriminant validity of the DOS in a large consecutive help-seeking sample at CAP clinics by comparing diagnosis-specific DOS scores between children with and without the diagnosis-specific disorder, and by using receiver operating characteristic (ROC) to examine the screening efficiency of the DOS. Secondary aims were to examine gender differences in mean scores as discovered in our

previous papers using same data [18, 19] and to evaluate the incremental validity of the DOS by combining data from multiple informants.

## Method

### Participants

In all, we included 307 CAP outpatients who consecutively sought treatment at four CAP clinics in southern Sweden from January 2010 to March 2013. Further information can be retrieved from our previous publications on this sample [12, 18, 19]. Briefly, our exclusion criterion was insufficient proficiency in Swedish by the patient or the parent. Forty cases were discarded due to protocol violations in the Schedule for Affective Disorders and Schizophrenia for School-Age Children–Present and Lifetime Version (K-SADS-PL) interview. One clinician used leading questions or did not ask both parent and child questions about all symptom areas and another clinician failed to sufficiently report data. The data from the remaining 267 cases are reported. These cases had a mean age of 12.1 (SD 3.2, range 6.1–17.8) years. The proportion of children 6–12 years was 57.7% (n = 154). There were slightly more boys (n = 150, 56.2%) than girls. The CBCL was filled out by 263 (98.5%) parents of these patients. Mothers' CBCL data were used in the parental CBCL in 240 (89.9%) cases; fathers' ratings were used only when those were the only data available (23 cases, 8.6%). The YSR was filled out by 139 (52.1%) young people, as it was only distributed to patients aged 11–17 years. Both YSR and CBCL ratings were available for 137 (51.3%) patients.

### Measures and procedures

A comprehensive description of measures and procedures can be found in a previous report [12]. Briefly, the semi-structured interview K-SADS-PL was used by resident MDs following a training program. The K-SADS interviews with both parents and patients yielded DSM-IV diagnoses, which were then further evaluated by using a Longitudinal Expert All Data (LEAD) process commonly viewed as the gold standard for evaluating semi-structured interviews. This process considers all information brought in through diagnostic procedures, the level of impairment, and the treatment outcome across a suitable period [39–42]. To be eligible for LEAD, the record should have covered at least six months of follow-up from the K-SADS-PL and included a minimum of three further visits or significant information from a teacher or an assessment by a senior clinician. In the LEAD work, the assessors had access to the K-SADS-PL interview as well as subsequent information from the medical records. All these data were retrieved by using a structured form. Thus, the re-evaluation of the K-SADS diagnoses was systematic and included oral reports and report forms from teachers and other informants, psychological assessments, and the outcome of pharmacological and psychological treatment. The observation time that yielded new diagnostic information was 1.2 (SD 0.6) years with a range of 0.1–3.1 years. For further information about the reliability of this process, see [12]. The LEAD procedure and clinical records were blind to the CBCL and YSR. The Ethical Review Board at Lund University approved the study. Patients aged 15 years and above and parents consented to the study in writing.

### ASEBA–Achenbach System and Empirically Based Assessment

The *Child Behavior Checklist (CBCL) for ages 6–18* [24] is a 120-item, parent-rated questionnaire designed to assess children's social competence and mental health problems. The equivalent self-rated questionnaire is The *Youth Self-Report (YSR) for ages 11–18*. Items on the two lists are rated on a 0–2 rating scale: 0 = not true; 1 = somewhat or sometimes true; 2 = very or

often true. Achenbach and Rescorla (2001) constructed a new scoring system for the CBCL and YSR scales, based on the DSM diagnostic criteria, the DSM-oriented scales, which will be used in the current study. The scales are affective problems, anxiety problems, attention-deficit/hyperactivity (ADHD) problems, oppositional defiant problems (ODD), and conduct problems (CD). We also examined OCD problems [38]. The internal consistency was as followed: affective (CBCL internal consistency ($\alpha = 0.82$, YSR $\alpha = 0.80$), anxiety (CBCL $\alpha = 0.82$, YSR $\alpha = 0.73$), ADHD (CBCL $\alpha = 0.84$, YSR $\alpha = 0.76$), ODD (CBCL $\alpha = 0.84$, YSR $\alpha = 0.61$), CD (CBCL $\alpha = 0.81$, YSR $\alpha = 0.82$), and OCD (CBCL $\alpha = 0.77$, YSR $\alpha = 0.76$).

## Statistics

T-tests were conducted to examine gender and diagnostic group differences on the DOS mean scores. Receiver operating characteristics (ROC) analyses were conducted to examine the concurrent validity of the CBCL/YSR DOSs versus a LEAD diagnosis [14, 43]. Generally, the area under the curve (AUC) is judged to represent low accuracy between 0.50 and 0.70, moderate accuracy between 0.70 and 0.90, and high accuracy above 0.90 [44]. The agreements between LEAD diagnoses and cut-off scores for the CBCL/YSR DOSs were also evaluated by using the Kappa statistic: poor agreement = less than 0.20; fair agreement = 0.20–0.40; moderate agreement = 0.40–0.60; good agreement = 0.60–0.80; and very good agreement = 0.80–1.00 [45]. We also conducted series of multivariate logistic regression analyses to evaluate the concurrent and discriminant validity of the CBCL/YSR DOSs. In addition, sequential logistic regression analyses were conducted to examine whether adding an informant (child or parent) would increase how accurately children with a disorder could be identified based on the relevant CBCL/YSR DOS. Sequential logistic regression was only used for participants 11 years or older since YSR was not administered to younger participants.

## Results

### Sample characteristics

The frequency of psychiatric disorders for the total sample and by gender is displayed in Table 1. The most prevalent disorders were ADHD (53%), anxiety disorders (36%), and depressive disorders (29%) while least prevalent were OCD (5%), and conduct disorders (4%).

### Disorder-specific and gender differences

Table 2 displays the means and standard deviations (SDs) for the CBCL and YSR for all DOSs. Participants diagnosed with a specific disorder (e.g., any depressive disorder) scored

**Table 1. The frequency of psychiatric disorders in the outpatient sample (n = 267).**

| Mental disorders | Boys | | Girls | | Total | |
|---|---|---|---|---|---|---|
| | n | % | n | % | n | % |
| Any depressive disorder | 40 | 26.7 | 40 | 34.2 | 80 | 30.0 |
| Anxiety disorders | 48 | 32.0 | 48 | 41.0 | 96 | 36.0 |
| ADHD | 91 | 60.7 | 51 | 43.6 | 142 | 53.2 |
| ODD | 34 | 22.7 | 28 | 23.9 | 62 | 23.2 |
| CD | 9 | 6.0 | 3 | 2.6 | 12 | 4.5 |
| OCD | 7 | 4.7 | 5 | 4.3 | 12 | 4.5 |

Any depressive disorder: Major Depressive Disorder, Dysthymic Disorder, or Depressive Disorder NOS.

Anxiety disorders: Generalized Anxiety Disorder, Separation Anxiety Disorder, Social Anxiety Disorder, Panic Disorder, Agoraphobia, and Specific Phobia.

**Table 2. Means, standard deviations, and independent t-test as per diagnostic group for CBCL and YSR and boys and girls.**

| Scale/subscale | All M (SD) | Disorder | | | | | Gender | | |
| | | Diagnosis-specific disorder present M (SD) | n | Disorder absent M (SD) | n | t-test | Boys M (SD) | Girls M (SD) | t-test |
| --- | --- | --- | --- | --- | --- | --- | --- | --- | --- |
| **CBCL** | n = 263 | | | | | | n = 146 | n = 117 | |
| **Affective** | 5.95 (3.40) | 8.86 (4.61) | 77 | 4.75 (3.96) | 186 | -7.283*** | 5.73 (4,25) | 6.24 (4,91) | -0.908 |
| **Anxiety** | 4.08 (3.30) | 5.85 (3.31) | 95 | 3.08 (2.86) | 168 | -7.125*** | 3.77 (3,27) | 4.47 (3,32) | -1.705 |
| **ADHD** | 6.39 (3.91) | 8.30 (3.46) | 139 | 4.24 (3.23) | 124 | -9.807*** | 6.91 (3,83) | 5.74 (3,93) | 2.445* |
| **ODD** | 4.80 (2.82) | 7.05 (2.06) | 61 | 4.12 (2.67) | 202 | -9.030*** | 4.84 (2,92) | 4.76 (2,71) | 0.213 |
| **CD** | 5.60 (4.85) | 14.45 (4.16) | 11 | 5.21 (4.51) | 252 | -6.678*** | 6.09 (4,90) | 4.99 (4,74) | 1.838 |
| **OCD** | 3.06 (2.86) | 8.58 (3.70) | 12 | 2.80 (2.53) | 252 | -7.547*** | 2.87 (2,87) | 3.31 (2,83) | -1.239 |
| **YSR** | n = 139 | | | | | | n = 66 | n = 73 | |
| **Affective** | 7.73 (5.07) | 10.20 (5.25) | 61 | 5.81 (4.00) | 78 | -5.416*** | 6.67 (4,84) | 8.70 (5,11) | -2.401* |
| **Anxiety** | 4.53 (3.24) | 6.19 (3.47) | 48 | 3.65 (2.74) | 91 | -4.727*** | 3.68 (3,15) | 5.29 (3,14) | -3.006** |
| **ADHD** | 6.09 (3.23) | 7.42 (3.24) | 60 | 5.09 (2.86) | 79 | -4.488*** | 6.05 (3,32) | 6.14 (3,17) | -0.166 |
| **ODD** | 4.71 (2.18) | 5.27 (1.76) | 26 | 4.58 (2.25) | 113 | -1.452 | 4.53 (2,14) | 4.88 (2,22) | -0.936 |
| **CD** | 4.94 (4.13) | 11.00 (5.90) | 8 | 4.57 (3.72) | 131 | -4.576*** | 5.03 (4,26) | 4.86 (4,03) | 0.238 |
| **OCD** | 4.58 (3.40) | 6.75 (4.37) | 8 | 4.45 (3.31) | 131 | -1.872 | 3.79 (2,99) | 5.30 (3,61) | -2.702** |

CBCL subscales explanations.

Significant differences between disorder diagnosed and gender: *** = p < .001, ** = p < .01 and * = p < .05.

CBCL = Child Behavior Checklist.

YSR = Youth-Self Report.

significantly higher on the corresponding DOS (e.g., affective) compared with participants without a specific disorder. However, we did not find any significant differences between participants with or without a diagnosis for YSR ODD and OCD subscale. We observed gender-specific differences for the CBCL ADHD scale, where parents scored significantly higher for boys than for girls. On the contrary, girls scored significantly higher than boys on the YSR affective, anxiety, and OCD scales.

## Diagnostic efficiency

First, we conducted a series of ROC analyses to evaluate how efficiently the DOSs predicted the presence of a corresponding LEAD diagnosis (Table 3). All predictions except YSR ODD and OCD were significant. We observed that CBCL CD predicted a diagnosis of CD with high accuracy. We observed moderate accuracy for the other DOSs in predicting the presence of their corresponding LEAD diagnoses.

Second, we selected the most efficient cut-off scores to equally minimize the false-positive and false-negative results by establishing maximizing efficiency κ(0.5) [29, 30]. Then, we evaluated the sensitivity and specificity of these cut-off scores (Table 3). For the CBCL and YSR affective scales, the Kappa [κ(0.5)] showed moderate agreement with their corresponding LEAD (any depressive disorder) diagnoses. The same was true for CBCL ADHD and OCD. All the other agreements were fair, except for YSR ODD and OCD, which showed poor agreement. Sensitivity ranged from 50% for YSR OCD to 81% for CBCL ADHD. The corresponding specificity ranged from 70% for the CBCL affective DOS, CBCL/YSR ADHD, and YSR ODD to 95% for CBCL CD (Table 3). More detailed results of the ROCs can be found in supplemental tables (see S1 File) with a presentation of each cutoff from 90% sensitivity to 90% specificity with kappa, positive and negative diagnostic likelihood ratio, and positive and negative predictive values.

**Table 3. Psychometric properties for the CBCL and YSR versus a LEAD diagnosis.**

|  | AUC (95% CI) | P | Cut-off | Sensitivity % | Specificity % | Kappa |
|---|---|---|---|---|---|---|
| CBCL Affective -> Any depression | .77 (.71, .82) | < .001 | ≥7 | 75 | 70 | .40 |
| CBCL Anxiety–> Any anxiety | .75 (.69, .80) | < .001 | ≥6 | 52 | 82 | .35 |
| CBCL ADHD -> ADHD | .81 (.75, .85) | < .001 | ≥6 | 81 | 70 | .51 |
| CBCL ODD -> ODD | .80 (.75, .85) | < .001 | ≥8 | 52 | 87 | .40 |
| CBCL CD -> CD | .93 (.89, .96) | < .001 | ≥14 | 55 | 95 | .38 |
| CBCL OCD -> OCD | .89 (.85, .92) | < .001 | ≥8 | 75 | 94 | .48 |
| YSR Affective -> Any depression | .74 (.66, .81) | < .001 | ≥9 | 67 | 76 | .43 |
| YSR Anxiety–> Any anxiety | .72 (.63, .78) | < .001 | ≥6 | 60 | 75 | .34 |
| YSR ADHD -> ADHD | .71 (.63, .78) | < .001 | ≥7 | 65 | 70 | .34 |
| YSR ODD -> ODD | .61 (.52, .69) | .060 | ≥6 | 54 | 70 | .18 |
| YSR CD -> CD | .84 (.77, .90) | < .001 | ≥9 | 75 | 85 | .29 |
| YSR OCD -> OCD | .66 (.57, .74) | .174 | ≥9 | 50 | 85 | .19 |

AUC = Area under the curve.

CBCL = Child Behavior Checklist.

YSR = Youth Self-Report.

## Concurrent and discriminant validity

We also conducted a series of multivariate logistic regression analyses to evaluate the concurrent and discriminant validity of each subscale. Thus, we aimed to verify whether only the corresponding subscale of the DOS is associated with particular LEAD diagnoses compared to the other subscales (Table 4). The odds ratios (ORs) showed that the CBCL's affective, anxiety, and ADHD, and ODD scales all predict the presence of their corresponding LEAD diagnoses.

**Table 4. Convergent/Divergent validity of the CBCL and YSR DOS versus LEAD diagnoses using multivariate logistic regression where the odds ratio (OR) refers to the likelihood of a diagnosis for every additional score point on each DOS.**

| CBCL | Depression OR (95% CI) p | Anxiety OR (95% CI) p | ADHD OR (95% CI) p | ODD OR (95% CI) p |
|---|---|---|---|---|
| $\chi^2$, p | 73.044, p < .001 | 69.678 p < .001 | 96.418 p < .001 | 64.383 p < .001 |
| CBCL Depression | 1.32 (1.20, 1.46) < .001 | 0.99 (.91, 1.07) .737 | 0.91 (0.83, 0.99) .024 | 0.93 (0.84, 1.02) .124 |
| CBCL Anxiety | 0.87 (0.75, 1.00) .054 | 1.52 (1.31, 1.77) < .001 | 0.89 (0.77, 1.02) .104 | 0.90 (0.77, 1.06) .217 |
| ADHD | 0.81 (0.73, 0.91) < .001 | 0.92 (0.83, 1.01) .84 | 1.46 (1.30, 1.64) < .001 | 0.96 (0.86, 1.07) .469 |
| ODD | 1.00 (0.84, 1.19) .997 | 0.99 (0.84, 1.17) .906 | 1.11 (0.94, 1.31) .223 | 1.67 (1.34, 2.08) < .001 |
| CD | 1.01 (0.90, 1.12) .905 | 0.14 (0.82, 1.02) .109 | 0.95 (0.86, 1.06) .354 | 1.03 (0.92, 1.14) .641 |
| OCD | 1.11 (0.94, 1.30) .223 | 0.85 (0.73, 1.00) .045 | 1.06 (0.90, 1.24) .501 | 1.06 (0.88, 1.28) .519 |
| YSR | Depression OR (95% CI) P | Anxiety OR (95% CI) p | ADHD OR (95% CI) p | ODD OR (95% CI) p |
| $\chi^2$, p | 50.829, p < .001 | 42.173 p < .001 | 37.316 p < .001 | 13.802 p = .032 |
| YSR Depression | 1.40 (1.21, 1.62) < .001 | 1.05 (0.92, 1.19) .472 | 0.83 (0.73, 0.95) .005 | 0.91 (0.78, 1.05) .196 |
| YSR Anxiety | 0.91 (0.75, 1.10) .334 | 1.42 (1.16, 1.75) .001 | 1.10 (0.91, 1.33) .329 | 1.06 (0.85, 1.32) .630 |
| ADHD | 0.72 (0.59, 0.88) .001 | 0.93 (.78, 1.11) .419 | 1.45 (1.22, 1.74) < .001 | 1.08 (0.90, 1.28) .415 |
| ODD | 0.72 (0.55, 0.95) .018 | .95 (.73, 1.25) .722 | 1.01 (0.79, 1.30) .914 | 1.21 (0.90, 1.63) .202 |
| CD | 1.21 (1.04, 1.40) .011 | 0.79 (0.67, 0.95) .011 | 1.05 (0.92, 1.21) .481 | 1.03 (0.88, 1.20) .730 |
| OCD | 1.04 (0.86, 1.26) .681 | 0.96 (0.79, 1.16) .671 | 0.91 (0.75, 1.09) .289 | 0.82 (0.65, 1.03) .083 |

CBCL = Child Behavior Checklist.

YSR = Youth Self-Report.

DOS = DSM oriented scale.

However, the ADHD DOS also significantly but negatively predicted the presence of a LEAD depression diagnosis. Likewise, the CBCL affective scale also significantly negatively predicted the presence of a LEAD ADHD diagnosis.

The YSR affective, anxiety, and ADHD DOSs predicted their corresponding LEAD diagnoses. However, YSR ODD did not predict the presence of the ODD diagnosis. Like the CBCL findings, we observed that the YSR ADHD and CD DOSs negatively predicted LEAD (any depression). Similarly, the YSR CD predicted LEAD anxiety. We did not analyze the data for CD or OCD due to too few diagnoses.

### Incremental validity

We evaluated the possible benefit of adding the DOS child report (YSR) to the parent report (CBCL) and vice versa in predicting LEAD diagnoses. We used a sequential logistic regression analysis to evaluate whether the DOSs would predict the presence of a depressive disorder, anxiety disorder, ADHD, ODD, CD, and OCD. First, we entered the parent report and then added the adolescent report. Second, we started with the adolescent report and then added the parent report. In this way, we evaluated the unique contribution of each informant to the other (Table 5). We found good goodness-of-fit values for all analyses (Hosmer–Lemeshow $p > 0.05$). For the affective scale, in the single variable models, both the YSR and the CBCL DOSs predicted the presence of depressive disorders (OR = 6.54 for YSR and OR = 5.29 for CBCL), explaining 24% of the variance ($R^2$). We observed significant benefits of adding the CBCL to the YSR ($\Delta\chi^2 = 16.172$, $p < 0.001$) and vice versa ($\Delta\chi^2 = 16.113$, $p < 0.001$). In the final model, both the CBCL and the YSR predicted the presence of depressive disorders (Table 5), explaining 36% of the variance. The DOS for anxiety predicted the presence of any anxiety disorder (OR = 4.88 for CBCL and OR = 4.56 for YSR), explaining 16% of the variance in the single variable models. Both scales demonstrated significant benefits when added to each other ($\Delta\chi^2 = 9.422$, $p < 0.05$ for adding the CBCL and $\Delta\chi^2 = 9.017$, $p < 0.05$ for adding the YSR) explaining 24% of the variance. The DOS for ADHD also predicted the presence of ADHD (OR = 8.18 for CBCL and OR = 4.39 for YSR explaining 28% and 16% of variance) in single variable models. Both scales showed significant benefits of adding an informant ($\Delta\chi^2 = 24.40$, $p < 0.001$ for adding the CBCL and $\Delta\chi^2 = 9.19$, $p < 0.05$ for adding the YSR) explaining 35% of the variance. We observed a significant OR (OR = 7.41 for CBCL and OR = 2.76 for YSR) when predicting ODD in the one informant (variable) model. However, only the CBCL carried significant benefits when added to the YSR ($\Delta\chi^2 = 11.435$, $p < 0.001$). Both scales had significant ORs when predicting the presence of CD (OR = 15.13 for CBCL and OR = 16.35 for YSR), but only the YSR carried significant benefits when added to the CBCL ($\Delta\chi^2 = 4.91$, $p < 0.05$). Both scales predicted the presence of OCD (OR = 52.29 for CBCL and OR = 5.79 for YSR) but only the CBCL carried significant benefits when added to the YSR ($\Delta\chi^2 = 17.74$, $p < 0.001$).

### Discussion

In the current study, we evaluated the concurrent and incremental validity of the CBCL and YSR DOSs with several DSMs internalizing and externalizing diagnoses based on the LEAD gold standard [12, 39]. This is the first study to evaluate the incremental validity of the CBCL added to YSR DOSs and vice versa.

In this sample of newly referred child and adolescent psychiatric outpatients, the concurrent validity of the parent reports (CBCL DOSs) showed moderate accuracy in predicting the presence of the corresponding disorder (AUC 0.75–0.89) while CD DOS predicted the presence of CD in the sample with high accuracy (AUC = 0.93). The child reports (YSR DOS) predicted the

**Table 5. Sequential logistic regression to test the effects of child and parent report on the DOS scales (using the most optimal cut-off scores) for the prediction of LEAD diagnoses.**

| LEAD diagnosis | Scale | OR (95%) | Wald | Full model $\chi^2$ | Full model adding an extra report $\chi^2$ | $R^2$ |
|---|---|---|---|---|---|---|
| Any depression Only one informant | CBCL | 6.98 (3.14, 15.51) | 22.762*** | 26.579*** | | .24 |
| | YSR | 6.54 (3.08, 13.87) | 23.946*** | 26.520*** | | .24 |
| Any depression Adding another informant | Add parent- to youth-report | 5.29 (2.27, 12.35) | 14.855*** | 42.692*** | Δ16.172*** | .36 |
| | Add youth—to parent-report | 4.97 (2.22, 11.09) | 15.329*** | 42.692*** | Δ16.113*** | |
| Any anxiety Only one informant | CBCL | 4.88 (2.34, 10.63) | 15.904*** | 16.680*** | | .16 |
| | YSR | 4.56 (2.14, 9.69) | 15.481*** | 16.276*** | | .16 |
| Any anxiety Adding another informant | Add parent- to youth-report | 3.61 (1.59, 8.20) | 9.354* | 25.698*** | Δ9.422* | .24 |
| | Add youth—to parent-report | 3.39 (1.52, 7.52) | 8.970** | 25.698*** | Δ9.017* | |
| ADHD Only one informant | CBCL | 8.18 (3.76, 17.79) | 28.147*** | 32.399*** | | .28 |
| | YSR | 4.39 (2.13, 9.04) | 16.099*** | 17.186*** | | .16 |
| ADHD Adding another informant | Add parent- to youth-report | 6.89 (3.08, 15.41) | 22.133*** | 41.587*** | Δ24.401*** | .35 |
| | Add youth—to parent-report | 3.39 (1.52, 7.52) | 8.974*** | 41.587*** | Δ9.188* | |
| ODD Only one informant | CBCL | 7.41 (2.69, 20.41) | 14.990*** | 14.736*** | | .16 |
| | YSR | 2.76 (1.15, 6.59) | 5.200* | 5.206* | | .06 |
| ODD Adding another informant | Add parent- to youth-report | 6.17 (2.17, 17.56) | 11.649*** | 16.641*** | Δ11.435*** | .18 |
| | Add youth—to parent-report | 1.96 (0.76, 5.04) | 1.942 | 16.641*** | Δ1.904 | |
| CD Only one informant | CBCL | 15.13 (3.18, 71.96) | 11.652*** | 10.289*** | | .20 |
| | YSR | 16.35 (3.08, 86.84) | 10.757*** | 12.852*** | | .25 |
| CD Adding another informant | Add parent- to youth-report | 4.10 (0.66, 25.38) | 2.299 | 15.197*** | Δ2.345 | .29 |
| | Add youth—to parent-report | 8.83 (1.29, 60.39) | 4.929* | 15.197*** | Δ4.908* | |
| OCD Only one informant | CBCL | 52.29 (8.89, 307.69) | 19.145*** | 22.551*** | | .42 |
| | YSR | 5.79 (1.33, 25.15) | 5.490* | 5.061* | | .10 |
| OCD Adding another informant | Add parent- to youth-report | 43.42 (6.49, 290.34) | 15.128*** | 22.800*** | Δ17.739*** | .43 |
| | Add youth—to parent-report | 1.62 (0.25, 10.57) | 0.254 | 22.800*** | Δ0.25 | |

CBCL (parent-report) = Child Behavior Checklist.

YSR (youth report) = Youth Self-Report.

Only one informant = univariate models.

Adding another informant = multivariate model.

corresponding LEAD-disorder with moderate accuracy. However, the accuracy of the youth ODD and OCD DOSs was low and not significant as opposed to the corresponding parent report. The scales also showed incremental validity when added to each other. However, adding the child as an informant did not increase diagnostic efficiency for ODD and OCD.

The low accuracy for the YSR ODD subscale is at odds with previous studies examining youth in the general population [30] or incarcerated adolescents [46]. Further, the YSR ODD scale had weak internal consistency (α = 0.61), supporting the inadequacy of this subscale in a clinical population. The diagnostic efficiency of the self-report (YSR) OCD scale has not been investigated previously. The low accuracy of the YSR OCD subscale is in line with studies of other self-report instruments for obsessive and compulsive symptoms in young people [47].

Cut-off values were chosen based on maximizing efficiency. We found moderate agreement between our cut-offs and LEAD diagnoses for the affective, ADHD, ODD, and OCD CBCLs (Kappa 0.40–0.51) and just slightly below moderate agreement for anxiety and CD (0.35, 0.38). All these cut-off values rendered acceptable sensitivity and specificity (e.g., affective scale: 75% sensitivity and 70% specificity) for screening in a clinical setting. The Kappa for the YSR scales showed moderate agreement with any depression but only fair agreement with anxiety, ADHD, and CD. However, we found poor agreement between the YSR OCD and ODD sub-scale and the corresponding LEAD diagnosis, reflecting the low AUC levels. Thus, most cut-off scores (especially the affective DOS (for both CBCL and YSR) and CBCL ADHD and OCD scales from the ROC analyses (based on the point where both sensitivity and specificity are optimal) can be used with confidence given that the sample is similar to the sample in our analyses.

We found clear evidence for both concurrent and discriminant validity of the DOSs for anxiety (CBCL and YSR) and for ODD (CBCL). Surprisingly, both affective and ADHD sub-scales (CBCL and YSR) predicted but also inversely predicted the presence of any depression or any ADHD. It is remarkable that both the CBCL and the YSR DOSs indicated a lower chance of depression with a high score on ADHD and vice versa despite the established comorbidity between these disorders. However, in this enriched clinical sample, patients with depression had clinically important comorbidity with ADHD but still a significantly lower rate of ADHD than those without depression (35% vs. 61%, p<0.001). We are not aware of any previous studies that have investigated the divergent validity of the DOS in a similar manner. The prevalence of both ADHD and any depressive disorder was high in this sample and the majority of the young people had at least one comorbid disorder [12], thus reflecting the clinical situation in a true manner and making screening and differential diagnostics more complicated.

Overall, we found good evidence that adding the parent as an informant, or vice versa, increases diagnostic precision. This is in line with a study of screening for depression with Mood and Feelings Questionnaire (MFQ), where a combination of parent and patient ratings was better than either rating alone [48]. When data has been analyzed separately across gender, it shows a significant contribution for adding parent ratings for adolescent girls but surprisingly not for boys [19], which would be important to examine further.

However, adding the child as the informant to information from parents does not increase diagnostic efficiency for the ODD and OCD DOSs, which corresponds to the findings of the ROC analyses. In addition, adding the parental information for CD DOS to information from the child does not increase diagnostic efficiency while for CD adding the child as an informant to the parent increases the diagnostic efficiency. This is not surprising as parents do not always have full knowledge about disruptive behaviors for adolescents.

The results also revealed that boys scored significantly higher on the CBCL ADHD. We did not find any gender differences in other CBCL DOS. Parents ratings for depression and anxiety were similar across gender while girls´ ratings were higher, which is in line with our findings from the Mood and Feelings Questionnaire (MFQ) [19] and Screen for Child Anxiety Related Emotional Disorders [18]. Our MFQ study also showed that parents and girls´ report correlated highly. However, the girls scored consequently higher, suggesting that girls express affective symptoms more markedly [19].

## Strengths and limitations

The main strength of this study was the large sample of participants from a specialized CAP clinical population. All patients were new referrals without prior contact with psychiatric

services. Thus, they had not received any prior psychiatric diagnosis, assessment or psychoe-ducation. This recruitment is ecologically suitable for testing the screening efficiency of the CBCL/YSR ahead of receiving a diagnosis. LEAD diagnoses were high quality, as they were based both on a semi-structured interview and on further clinical work-up and observations as well as expert consensus by two senior consultants (TI and HJ). The expert consensus work was independent of the ASEBA scores, as no information from the scales was included in the clinical records. There were adequate numbers for ADHD (n = 60), anxiety disorders (n = 48) and depression (n = 61) on the YSR self-report for analyzing concurrent and incremental validity of the DOSs.

However, there were some limitations as well. First, although this was a sizable study, the number of patients in some diagnostic groups was small. For instance, we had only 11 partici-pants with CD and 12 with OCD limiting our analysis strategy, especially for convergent and divergent validity using logistic regression. Second, using LEAD diagnoses based on enhancing K-SADS with information from clinical records is still at risk of including spurious variation.

## Conclusion

In a child and adolescent outpatient psychiatric setting, the subscales of CBCL and YSR for ADHD, anxiety disorders, depression, and conduct disorders and the CBCL subscales for ODD and OCD can be used for screening or for enhancing diagnostic assessment. Adding self-report to parent-report and vice versa improves the prediction and is recommended for youths. YSR self-report for OCD and ODD should not be used.

## Supporting information

**S1 File.**
(DOCX)

**S1 Data.**
(SAV)

## Author Contributions

**Conceptualization:** Gudmundur Skarphedinsson, Håkan Jarbin, Markus Andersson, Tord Ivarsson.

**Formal analysis:** Gudmundur Skarphedinsson.

**Funding acquisition:** Håkan Jarbin.

**Investigation:** Gudmundur Skarphedinsson, Håkan Jarbin, Markus Andersson, Tord Ivarsson.

**Methodology:** Håkan Jarbin, Markus Andersson, Tord Ivarsson.

**Validation:** Gudmundur Skarphedinsson.

**Writing – original draft:** Gudmundur Skarphedinsson.

**Writing – review & editing:** Håkan Jarbin, Markus Andersson, Tord Ivarsson.

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
