## [Decision Letter · Decision Letter 0]

26 May 2021

PONE-D-21-10635

Diagnostic efficiency and validity of the DSM-oriented Child Behavior Checklist and Youth Self-Report scales in a clinical sample of Swedish youth

PLOS ONE

Dear Dr. Skarphedinsson,

Thank you for submitting your manuscript to PLOS ONE. After careful consideration, we feel that it has merit but does not fully meet PLOS ONE’s publication criteria as it currently stands. Therefore, we invite you to submit a revised version of the manuscript that addresses the points raised during the review process.

I was hoping to receive a second review of your work, but, unfortunately, it has not been received. I thank the reviewer for their attention to the manuscript. You will see that the reviewer noted a number of clarifications for your work. I concur with the reviewer that the work is done well. The supplementary material, particularly the tables showing the full set of AUC values across scale cutoff values was very helpful. However, I have some additional queries and comments that are intended to help move the work along.

The motivation for the work on the use of the DSM oriented scale (DOS) scores and diagnostic specificity is clear. However, there could be some additional details about the motivation to rely on youth and maternal reports, rather than other selection of informants.

In the initial paragraph of the Methods, you note: “The observation time that yielded new diagnostic information was 1.2 (SD 0.6) years with a range of 0.1–3.1 years.” It is not clear, at this point in the manuscript, what this information is about. Is this about using the LEAD process? If so, this should be integrated into the description.

Table 2 presents DOS between youth with and without diagnoses and between male and female youth. However, the comparisons of male and female youth are not motivated in the introduction or commented on in the Discussion. You have a choice in how to handle the inclusion/exclusion of these analyses. Please justify your decision; if these are retained be sure to justify their inclusion in the manuscript.

Tables 2 & 3 show associations between DOS scores and diagnoses as mean differences and AUC, respectively. These appear to be reparametrized estimates of the same quantity. Please describe how these are different. Moreover, it was not clear whether the models estimated in Table 4 included only a single predictor in each model. If so, then these ORs would be a third representation of the same information. Please clarify whether the models in Table 4 include one or more predictors in the same model.

In the models where youth report was added to parent report, in that order, were parent reports only included if they also had youth reports? If not, then the model R^2^ for the initial step would be based on different data than the model with both informants.

In the Discussion, some results are described as if there were direct tests of differences in magnitudes of association. However, the analyses, as presently communicated, are only showing whether there were significant or non-significant associations between DOS scores and diagnoses. This language should be addressed.

We look forward to receiving your revised manuscript.

Kind regards,

Thomas M. Olino

Academic Editor

PLOS ONE

Journal Requirements:

2. Please include your tables as part of your main manuscript and remove the individual files. Please note that supplementary tables should remain uploaded as separate "supporting information" files.

4. We noticed you have some minor occurrence of overlapping text with the following previous publication(s), which needs to be addressed:

- https://journals.plos.org/plosone/article?id=10.1371%2Fjournal.pone.0230623

- https://link.springer.com/article/10.1007%2Fs10578-017-0746-8

In your revision ensure you cite all your sources (including your own works), and quote or rephrase any duplicated text outside the methods section. Further consideration is dependent on these concerns being addressed.

Reviewers' comments:

Reviewer's Responses to Questions

**Comments to the Author**

1. Is the manuscript technically sound, and do the data support the conclusions?

Reviewer #1: Yes

2. Has the statistical analysis been performed appropriately and rigorously? 

Reviewer #1: Yes

3. Have the authors made all data underlying the findings in their manuscript fully available?

Reviewer #1: Yes

4. Is the manuscript presented in an intelligible fashion and written in standard English?

Reviewer #1: Yes

5. Review Comments to the Author

Reviewer #1: This is a study on the clinical properties of CBCL, with an excellent experimental design and an advanced analysis of the collected data. It is probably the most accurate CBCL study to date.

The CBCL is a widely used tool, with very high number of citations on PubMed, but there are not many studies on its clinical properties in relation to scaling according to DSM criteria. In the introduction (second page) the authors report these studies; probably for completeness it is appropriate to add the most recent published in Clin Child Psychol Psychiatry 2020;25:507-519. doi: 10.1177/1359104519895056.

The clinical analysis is particularly accurate according to the criteria of the Longitudinal Expert All Data (LEAD) procedure, even if, considering the variability over time of the clinical picture in children and adolescents, it is questionable whether a final evaluation at an average distance of many months may sometimes not exactly correspond to the clinical situation at the time of administration of the CBCL

In the "Diagnostic efficiency" section there is a repetition relating to YSR ODD and OCD.

In the “Concurrent and discriminant validity” section, please check what is written in relation to the YSR (last lines) in relation to the data in table 4.

As for the incremental validity, I wonder if adding CBCL to YSR and vice versa can increase the accuracy of that of the two which is already more accurate.

Some minor corrections in the tables.

Table 2 gender t-test YSR ODD -9.360 (probably wrong)

Decimal separator: sometimes is comma instead of dot.

6. PLOS authors have the option to publish the peer review history of their article (what does this mean?). If published, this will include your full peer review and any attached files.

Reviewer #1: No

---

## [Author Response · Author response to Decision Letter 0]

30 Jun 2021

Diagnostic efficiency and validity of the DSM-oriented Child Behavior Checklist and Youth Self-Report scales in a clinical sample of Swedish youth

Authors´responses to reviews

Editor

I was hoping to receive a second review of your work, but, unfortunately, it has not been received. I thank the reviewer for their attention to the manuscript. You will see that the reviewer noted a number of clarifications for your work. I concur with the reviewer that the work is done well. The supplementary material, particularly the tables showing the full set of AUC values across scale cutoff values was very helpful. However, I have some additional queries and comments that are intended to help move the work along.

Authors‘ response

Thank you.

Editor

The motivation for the work on the use of the DSM oriented scale (DOS) scores and diagnostic specificity is clear. 

However, there could be some additional details about the motivation to rely on youth and maternal reports, rather than other selection of informants.

Authors‘ response

Unfortunately, we only had access to youth and maternal reports. However, we acknowledge that it is a strength to add an additional informant such as father or teacher.

Editor

In the initial paragraph of the Methods, you note: “The observation time that yielded new diagnostic information was 1.2 (SD 0.6) years with a range of 0.1–3.1 years.” It is not clear, at this point in the manuscript, what this information is about. Is this about using the LEAD process? If so, this should be integrated into the description.

Authors‘ response

Thank you. We have moved this sentence to the measures and procedures section where the LEAD process is described. 

Editor

Table 2 presents DOS between youth with and without diagnoses and between male and female youth. However, the comparisons of male and female youth are not motivated in the introduction or commented on in the Discussion. You have a choice in how to handle the inclusion/exclusion of these analyses. Please justify your decision; if these are retained be sure to justify their inclusion in the manuscript.

Authors‘ response

Thank you.

We have added a sentence in the aim section

„by comparing diagnosis-specific DOS scores between children with and without the diagnosis-specific disorder“.

and

„Our aim was also to examine potential gender differences in mean scores as discovered in our previous papers using same data {Ivarsson et al., 2017, #2921}{Jarbin et al., 2020, #5849}.“

We have also added a paragraph in the discussion

“The results also revealed that boys scored significantly higher on the CBCL ADHD. We did not find any gender differences in other CBCL DOS. Parents ratings for depression and anxiety were similar across gender while girls´ ratings were higher, which is in line with our findings from the Mood and Feelings Questionnaire (MFQ) {Jarbin et al., 2020, #5849} and Screen for Child Anxiety Related Emotional Disorders {Ivarsson et al., 2017, #2921}. Our MFQ study also showed that parents and girls´ report correlated highly. However, the girls scored consequently higher, suggesting that girls express affective symptoms more markedly {Jarbin et al., 2020, #5849}. “

Editor

Tables 2 & 3 show associations between DOS scores and diagnoses as mean differences and AUC, respectively. These appear to be reparametrized estimates of the same quantity. Please describe how these are different. 

Authors‘ response

Thank you. We acknowledge that a part of table 2 (M and SD of diagnosis-specific disorder present/absent) is a reparameterization of table 3. However, we consider both tables to be valuable for our readers (clinicians and researchers). Table 2 added value is to compare M and SDs in this sample to clinical or community samples. Table 3 added value is to present AUCs and the most efficient cutoffs. Please, let us know if you need further elaboration or if changes are suggested.

Editor

Moreover, it was not clear whether the models estimated in Table 4 included only a single predictor in each model. If so, then these ORs would be a third representation of the same information. Please clarify whether the models in Table 4 include one or more predictors in the same model.

Authors‘ response

Thank you for noting that. We used multivariate logistic regression, all the DOS scales were used as independent variables in each model to determine the specific association between disorder (e.g., depression) and diagnosis-specific scale (e.g., CBCL depression). We apologize for this misunderstanding. We have added “multivariate” in the title. We have also added “multivariate” in statistical analysis and in the results section. We acknowledge that the univariate models are representations of the same information. however, we deem it important as it might be important for our readers to compare the ORs between the univariate and multivariate models.

Editor

In the models where youth report was added to parent report, in that order, were parent reports only included if they also had youth reports? If not, then the model R2 for the initial step would be based on different data than the model with both informants.

Authors‘ response

Thank you for noting that. Yes, we made sure to include only parent reports if they also had youth reports.

Editor

In the Discussion, some results are described as if there were direct tests of differences in magnitudes of association. However, the analyses, as presently communicated, are only showing whether there were significant or non-significant associations between DOS scores and diagnoses. This language should be addressed.

Authors‘ response

Thank you. Language is changed in Discussion, 2nd paragraph, line 6. 

Editor

Authors‘ response

(remember 

(1) respones

(2) track changes

(3) CLEAN

If applicable, we recommend that you deposit your laboratory protocols in protocols.io to enhance the reproducibility of your results. Protocols.io assigns your protocol its own identifier (DOI) so that it can be cited independently in the future. For instructions see: http://journals.plos.org/plosone/s/submission-guidelines#loc-laboratory-protocols. Additionally, PLOSONE offers an option for publishing peer-reviewed Lab Protocol articles, which describe protocols hosted on protocols.io. Read more information on sharing protocols at https://plos.org/protocols?utm_medium=editorial-email&utm_source=authorletters&utm_campaign=protocols.

We look forward to receiving your revised manuscript.

Kind regards,

Thomas M. Olino

Academic Editor

PLOS ONE

Journal Requirements:

1. Please ensure that your manuscript meets PLOSONE's style requirements, including those for file naming. The PLOS ONE style templates can be found at

https://journals.plos.org/plosone/s/file?id=wjVg/PLOSOne_formatting_sample_main_body.pdfand

2. Please include your tables as part of your main manuscript and remove the individual files. Please note that supplementary tables should remain uploaded as separate "supporting information" files.

Authors‘ response

(Remember to add all tables to the Ms.)

Editor

4. We noticed you have some minor occurrence of overlapping text with the following previous publication(s), which needs to be addressed:

- https://journals.plos.org/plosone/article?id=10.1371%2Fjournal.pone.0230623

- https://link.springer.com/article/10.1007%2Fs10578-017-0746-8

In your revision ensure you cite all your sources (including your own works), and quote or rephrase any duplicated text outside the methods section. Further consideration is dependent on these concerns being addressed.

 Authors‘ response

Thank you for noting that. We are the co-authors of both papers. We confirm that we only duplicated some parts of the method section.

Reviewers' comments:

Reviewer's Responses to Questions

Comments to the Author

1. Is the manuscript technically sound, and do the data support the conclusions?

Reviewer #1: Yes

2. Has the statistical analysis been performed appropriately and rigorously? 

Reviewer #1: Yes

3. Have the authors made all data underlying the findings in their manuscript fully available?

Reviewer #1: Yes

4. Is the manuscript presented in an intelligible fashion and written in standard English?

Reviewer #1: Yes

5. Review Comments to the Author

R1

Reviewer #1: This is a study on the clinical properties of CBCL, with an excellent experimental design and an advanced analysis of the collected data. It is probably the most accurate CBCL study to date.

The CBCL is a widely used tool, with very high number of citations on PubMed, but there are not many studies on its clinical properties in relation to scaling according to DSM criteria. In the introduction (second page) the authors report these studies; probably for completeness it is appropriate to add the most recent published in Clin Child Psychol Psychiatry 2020;25:507-519. doi: 10.1177/1359104519895056.

Authors‘ response

Thank you so much. We have added the study in our manuscript, when we list the studies that have examined the screening efficiency of the DOS depression, anxiety, ADHD, ODD, and CD. Great paper and interesting to read about the CABI.

R1

The clinical analysis is particularly accurate according to the criteria of the Longitudinal Expert All Data (LEAD) procedure, even if, considering the variability over time of the clinical picture in children and adolescents, it is questionable whether a final evaluation at an average distance of many months may sometimes not exactly correspond to the clinical situation at the time of administration of the CBCL

Authors‘ response

Thank you. We acknowledge, that LEAD diagnoses are not without problems when data over time is used. However, we want to cite the following paper (by the same research group): http://dx.doi.org/10.1080/08039488.2016.1276622. In this paper further description on our LEAD method can be accessed. the LEAD diagnoses were based on KSADS data and a minimum of three visits after the KSADS or of significant new information. KSADS diagnoses (at the time of intake) were compared to LEAD diagnoses. The results showed excellent agreement for most diagnoses. For instance, any depression kappa = 0.91, any anxiety disorders kappa = 0.94, any ADHD kappa 0.80. Among these diagnoses, the most notable information arriving after the KSADS-PL was information in rating scales and personal communication from teachers supporting a diagnosis of ADHD. Parents and patients were, at times, reducing their initial description of ADHD and functional impact but agreed later on when school had presented more support and they also were less reluctant of receiving the diagnosis. As ADHD has a chronic fluctuating course, we believe that the later additional information was valuable and correct. There were very few changes of affective diagnosis but in a rare case, the depression was covered by a panic disorder but elicited later on and clearly with an onset before the time of the CBCL/YSR. 

R1

In the "Diagnostic efficiency" section there is a repetition relating to YSR ODD and OCD.

Authors‘ response

Thank you so much. We have changed accordingly by deleting the second sentence about ODD and OCD.

R1

In the “Concurrent and discriminant validity” section, please check what is written in relation to the YSR (last lines) in relation to the data in table 4.

Authors‘ response

Thank you again. We have modified the sentences according to the results in table 4.

R1

As for the incremental validity, I wonder if adding CBCL to YSR and vice versa can increase the accuracy of that of the two which is already more accurate.

Authors‘ response

Again, thank you. We understand that if we add CBCL DOS (e.g., depression) to YSR DOS (e.g., depression) the accuracy of predicting LEAD depression increases and vice versa. Yes, that´s exactly what happens :) if the delta (�) (i.e., increase in �2) is significant than adding CBCL to YSR and vice versa increases tha accuracy. Except for adding YSR ODD to CBCL ODD. Adding CBCL CD to YSR CD, and Adding YSR OCD to CBCL OCD.

R1

Some minor corrections in the tables.

Table 2 gender t-test YSR ODD -9.360 (probably wrong)

Authors‘ response

Wonderful, thank you so much for your sharp eyes. The correct t is 0.936. We missed one decimal.

R1

Decimal separator: sometimes is comma instead of dot.

Authors‘ response

Thank you and apologies. We have changed from comma to dot accordingly.

6. PLOS authors have the option to publish the peer review history of their article (what does this mean?). If published, this will include your full peer review and any attached files.

Do you want your identity to be public for this peer review? For information about this choice, including consent withdrawal, please see our Privacy Policy.

Reviewer #1: No

While revising your submission, please upload your figure files to the Preflight Analysis and Conversion Engine (PACE) digital diagnostic tool, https://pacev2.apexcovantage.com/. PACE helps ensure that figures meet PLOS requirements. To use PACE, you must first register as a user. Registration is free. Then, login and navigate to the UPLOAD tab, where you will find detailed instructions on how to use the tool. If you encounter any issues or have any questions when using PACE, please email PLOSat figures@plos.org. Please note that Supporting Information files do not need this step.

---

## [Editor Report · Decision Letter 1]

7 Jul 2021

Diagnostic efficiency and validity of the DSM-oriented Child Behavior Checklist and Youth Self-Report scales in a clinical sample of Swedish youth

PONE-D-21-10635R1

Dear Dr. Skarphedinsson,

We’re pleased to inform you that your manuscript has been judged scientifically suitable for publication and will be formally accepted for publication once it meets all outstanding technical requirements. Your responses were clear and the manuscript was clarified. 

Kind regards,

Thomas M. Olino

Academic Editor

PLOS ONE
---

## [Editor Report · Acceptance letter]

14 Jul 2021

PONE-D-21-10635R1 

Diagnostic efficiency and validity of the DSM-oriented Child Behavior Checklist and Youth Self-Report scales in a clinical sample of Swedish youth 

Dear Dr. Skarphedinsson:

I'm pleased to inform you that your manuscript has been deemed suitable for publication in PLOS ONE. Congratulations! Your manuscript is now with our production department. 

Kind regards, 

on behalf of

Dr. Thomas M. Olino 

Academic Editor

PLOS ONE